# A minisatellite-based MLVA for deciphering the global epidemiology of the bacterial cassava pathogen *Xanthomonas phaseoli* pv. *manihotis*

Leidy Rache[1¤], Laurence Blondin[2,3], Paula Diaz Tatis[4], Carolina Flores[3], Andrea Camargo[4], Moussa Kante[3], Issa Wonni[5], Camilo López[6], Boris Szurek[3], Stephane Dupas[1], Olivier Pruvost[7], Ralf Koebnik[3]*, Silvia Restrepo[1], Adriana Bernal[1]*, Christian Vernière[3,7†]

1 Department of Biological Sciences, Universidad de los Andes, Bogotá D.C., Colombia, 2 CIRAD, UMR PHIM, Montpellier, France, 3 Plant Health Institute of Montpellier (PHIM), Université Montpellier, IRD, CIRAD, INRAE, Institut Agro, Montpellier, France, 4 Facultad de Ciencias, Universidad Antonio Nariño, Bogotá, Colombia, 5 INERA, LMI Pathobios, Bobo Dioulasso, Burkina Faso, 6 Departamento de Biología, Universidad Nacional de Colombia, Bogotá, Colombia, 7 CIRAD, UMR PVBMT, Saint-Pierre, La Réunion, France

† Deceased.
¤ Current address: Escuela de Ciencias Biológicas, Universidad Pedagógica y Tecnológica de Colombia, Tunja, Colombia
* ralf.koebnik@ird.fr (RK); abernal@uniandes.edu.co (AB)

## Abstract

Cassava Bacterial Blight (CBB) is a destructive disease widely distributed in the different areas where this crop is grown. Populations studies have been performed at local and national scales revealing a geographical genetic structure with temporal variations. A global epidemiology analysis of its causal agent *Xanthomonas phaseoli* pv. *manihotis* (*Xpm*) is needed to better understand the expansion of the disease for improving the monitoring of CBB. We targeted new tandem repeat (TR) loci with large repeat units, i.e. minisatellites, that we multiplexed in a scheme of Multi-Locus Variable number of TR Analysis (MLVA-8). This genotyping scheme separated 31 multilocus haplotypes in three clusters of single-locus variants and a singleton within a worldwide collection of 93 *Xpm* strains isolated over a period of fifty years. The major MLVA-8 cluster 1 grouped strains originating from all countries, except the unique Chinese strain. On the contrary, all the *Xpm* strains genotyped using the previously developed MLVA-14 microsatellite scheme were separated as unique haplotypes. We further propose an MLVA-12 scheme which takes advantage of combining TR loci with different mutation rates: the eight minisatellites and four faster evolving microsatellite markers, for global epidemiological surveillance. This MLVA-12 scheme identified 78 haplotypes and separated most of the strains in groups of double-locus variants (DLV) supporting some phylogenetic relationships. DLV groups were subdivided into closely related clusters of strains most often sharing the same geographical origin and isolated over a short period, supporting epidemiological relationships. The main MLVA-12 DLV group#1 was composed by strains from South America and all the African strains. The MLVA-12

**Data Availability Statement:** All relevant data are within the paper and its supporting information files.

**Funding:** We thank the Faculty of Sciences from Universidad de los Andes-Bogotá, Colombia (INV-2021-128-2283), the UMR Interactions Plantes Microorganismes Environnement, and the Agropolis Foundation (project PAIX, grant no. 1 403-073), Montpellier, France, for financial support. The Ecos Nord programme supported inter-laboratory mobility (grant no. C15A01). Leidy Rache was supported by Colciencias with a doctoral fellowship (call No. 528, 2011). Olivier Pruvost acknowledges the European Union (ERDF contract GURDT I2016-1731-0006632) and Réunion regional council for support.The funders had no role in study design, data collection and analysis, decision to publish, or preparation of the manuscript

**Competing interests:** The authors have declared that no competing interests exist.

scheme combining both minisatellite and microsatellite loci with different discriminatory power is expected to increase the accuracy of the phylogenetic signal and to minimize the homoplasy effects. Further investigation of the global epidemiology of *Xpm* will be helpful for a better control of CBB worldwide.

## Introduction

Molecular genotyping of bacterial pathogens allows a comprehensive view of their epidemiology and evolution. The ability to differentiate genotypes at different spatial and temporal scales can be informative of the dispersion and micro or macroevolution of bacterial pathogens [1–3]. At a local scale, this approach can shed light on the microevolutionary processes within populations, and track back the spread of bacterial isolates and the associated pathways. At a long-term or global scale, it can for instance unravel the invasion routes and characterize some bacterial lineages associated to pathological or adaptive traits for epidemiological surveillance. Genotyping methods based on several molecular markers have been developed to analyze the genetic diversity of pathogenic bacteria. The type of marker should be selected according to the research question to be answered because each marker has different mutation rates i.e. different speed of evolution and stability [2–6]. Markers with a desired molecular clock can be selected according to the evolutionary scale under study.

Global epidemiology studies or international surveillance of pathogens require tools that can produce unambiguous data and are easily transferrable among different laboratories. Multilocus sequence typing (MLST) is a nucleotide sequence-based approach using markers that are conserved, stable, with relatively low evolution rates. It enables international comparison of isolates and global epidemiology [7]. However, some bacterial pathogens, the so-called genetically monomorphic bacteria [8], do not exhibit enough variability in their housekeeping genes making this approach insufficiently resolutive [8]. Various subtyping methods emerged with the advent of whole genome sequencing. In situations where large numbers of high-quality draft genomes can be obtained, single nucleotide polymorphism (SNP) typing is a powerful approach for epidemiological surveillance and phylogenetic analyses [8–11]. Technically simple and still robust systems are also necessary for routine epidemiological investigations in situations where an easy access to large sets of compete genomes is impossible to achieve. In such situations, Tandem Repeats (TRs) and Clustered Regularly Interspaced Short Palindromic Repeats (CRISPRs) have been mainly applied as typing tools with regard to their reliability, discriminatory power, transferability, ease of application and cost [12,13].

TRs are short motifs of DNA fragments repeated in tandem that can vary in number due to addition or deletion of motifs. Multiple loci variable number tandem repeat analysis (MLVA) targets multiple TR loci which allows a high discriminatory power if a sufficiently large number of TRs is targeted. Two classes have been proposed to separate these informative molecular markers: microsatellites with a repeat unit size < 10 bp and minisatellites with repeat units ≥ 10 bp [14,15]. The length of TR arrays is typically determined by capillary electrophoresis from multiplex involving fluorochrome-labeled primers but it can be scored unambiguously and from agarose gels especially for large repeat motifs making them applicable at a lower throughput in laboratories where a capillary electrophoresis genotyper is not available. Several microsatellite schemes have been developed for epidemiological analyses of plant pathogenic xanthomonads [16]. Minisatellite-based typing schemes targeting large repeat motifs

were also proposed as a portable approach for the global analysis of the molecular epidemiology of bacterial strains pathogenic to human or animals [15,17,18] or to plant crops [19].

The wide range of mutation rates offered by the different TR loci classes make them useful for examining genetic patterns at several evolutionary scales. While microsatellites are most suited for local epidemiology studies, minisatellites have been mostly used for global epidemiology, *i.e.* analysis of non-epidemiologically related isolates [17,19].

Mutational processes observed in VNTRs are mainly driven by DNA replication slippage (and to a lesser extent recombination) and the length or the number of the tandem repeat and the sequence of the repeat unit are factors that drive the mutation rate of TRs [14,20,21].

Cassava Bacteria Blight (CBB) is a very important disease caused by *Xanthomonas phaseoli* pv. *manihotis* (*Xpm*) [22,23]. This disease was first reported in Brazil in 1912 [24], and later noticed in Colombia in 1970 [25], Venezuela in 1971 and African and Asian countries since 1972 [26,27]. CBB might have been present in these countries for many years, but few reports described a bacterial leaf spot on cassava before the seventies. More recently, the disease was reported in South Pacific [28]. Population studies of the pathogen have been conducted in several Latin American countries, including Brazil, Venezuela [29,30] and a continuous monitoring of the populations has been performed in Colombia [31–36]. *Xpm* is transmitted at small scale by rain and wind within production areas. At larger scales between production areas and between different countries, contaminated plant material, i.e. stem cuttings, are responsible for bacterial dispersal and the carry-over to the following growing season [24].

Studies of the genetic diversity in *Xpm* populations have been mainly performed at a local level using Restriction Fragment Length Polymorphisms (RFLP) or Amplified Fragment Length Polymorphisms (AFLP), as well as whole genome sequencing [37]. In Colombia *Xpm* populations were genetically diverse and geographically structured [34,38]. Genetic variation was observed within fields with high levels of genotypic diversity and haplotype frequencies significantly differing over the years [33,38]. More recent studies in different regions of Colombia confirmed the high genetic diversity and the local and geographic structure of *Xpm* populations in these regions [35,36]. A geographic structure of *Xpm* populations was also described in Venezuela at a regional scale [29]. Similarly, a high population diversity has been observed in African countries [39].

A few large-scale analyses of the genetic diversity of *Xpm* populations have been performed. Analysis of a worldwide collection using RFLP led to the identification of clusters grouping strains that originated from different countries [26]. This study suggested that African strains belong to a single subclade of South American strains, whereas Asian strains belong to different groups. A recent study based on single nucleotide polymorphism analyses supports gene flow among *Xpm* strains representative of the worldwide diversity especially between Brazil and Colombia and between the South-American and the African continents [37]. A common ancestor among Brazil, Colombia and African strains was suggested as previously suspected [26].

Herein, we built on the versatility of TR markers in order to develop improved MLVA schemes with desirable characteristics for outbreak investigation or epidemiological surveillance of *Xpm*. An MLVA-scheme targeting microsatellites was recently proposed and evaluated on local populations [40]. Our aims in this study were (i) to develop new TR markers with larger repeat size, (ii) to evaluate the discriminatory power of these minisatellites on a worldwide collection of *Xpm* strains in comparison with the MLVA scheme based on microsatellites and (iii) to propose an MLVA scheme adapted for regional or international surveillance of *Xpm*.

## Materials and methods

### *Xpm* strain collection and DNA extraction

A collection of 93 *Xpm* strains isolated over several years from different cassava producing countries from South America (Colombia, Venezuela, Brazil, Argentina), Africa (Burkina Faso, Mali, Togo), Asia (Vietnam, and China) and New Zealand, (S1 Table) was used for genotyping. The genomic DNA was extracted from 24 hour-old cultures on an LPGA culture medium (Yeast extract 5 g L$^{-1}$, Peptone 5 g L$^{-1}$, glucose 5 g L$^{-1}$, agar 15 g L$^{-1}$, pH 7.2). DNA extraction was performed using the Wizard Ⓡ Genomic kit (Promega, Madison, WI, USA). The amount and purity of DNA was checked using NanoDrop technology (Thermoscientific, Illkirch, France).

### Genotyping using a new MLVA scheme targeting minisatellites

Twelve *Xpm* genomes, previously sequenced with Illumina [37], were selected, based on degree of fragmentation (least fragmented were included) and number of contigs (lower number of contigs were included). Eight strains from Brazil, two from Colombia and two from Uganda [40], were screened for VNTR minisatellite loci using the *Xanthomonas* utility website (http://bioinfo-web.mpl.ird.fr/xantho/utils/), setting the following parameters: 30 and 1000 bp for the total length, the unit length of the repeat in a range of 10 to 100 bp, similarity of repeats within the arrays between 80 and 100%.

The primers were designed over flanking regions up- and downstream of the repeat region as previously described [40]. Primers were tested by individual PCR and multiplex PCR using DNA from a sub-collection of *Xpm* strains: CIO1, CIO151, COL303, CFBP1851 (Colombia), VEN130 (Venezuela), SCI (Ivory Coast), D2-3 (Burkina Faso), Mali45, Mali30 (Mali). These strains were only used to test the quality and design of the primers in agarose gels.

Multiplex PCR was performed to amplify three loci in each multiplex using the QIAGEN$^{®}$ Multiplex PCR kit (Qiagen, Courtaboeuf, France) according to the manufacturers´ instructions with 40 to 50 ng of target in a total volume of 10 µl. The multiplex PCR conditions were as follows: pre-denaturation at 95˚C for 5 minutes, 30 cycles of denaturation at 95˚C for 30 seconds, annealing with a specific temperature per pool for 40 seconds (Table 1), extension at 72˚C for 2 minutes; and a final extension at 72˚C for 5 minutes. The amplicons were visualized by agarose gel electrophoresis (2%).

After the standardization of the multiplex PCR, the reverse primer in each primer pair was labelled using different fluorescent dyes to separate the PCR products by capillary electrophoresis (Table 1). PCR products were diluted 20 times (estimated from preliminary tests), and 1 µl of the dilution was mixed with HiDi formamide (9 µl) (Applied Biosystems, CA, USA), and GeneScan$^{TM}$ 600 LIZ$^{®}$ dye size standard v2.0 (0.5 µl) (Applied Biosystems, CA, USA). The detection was performed in an ABI 3500 Genetic Analyzer, and amplicon sizes were determined using GeneMapperⓇ (Applied Biosystems) software version 4.1. The strain CIO151 [41] was used as a reference strain in each test.

### *Xpm* genotyping using MLVA-14 targeting microsatellites

The strain collection was also genotyped using a MLVA scheme targeting 14 microsatellites, referred to as MLVA-14. This scheme is derived from MLVA-15 [40], but a single microsatellite, Xpm 2–20 (VNTR Xpm 1–20 nomenclature from [40]), was taken out because it did not amplify on a few strains from Venezuela or Africa (unpublished data). One primer from each pair in the PCR mix was labelled with one of the four different fluorescent dyes (6-FAM, NED,

**Table 1. PCR primers sequences and fluorophore dyes of the minisatellite loci used in the Multiplex PCR with the corresponding annealing temperature (AT).**

| Pool | VNTR locus | Forward sequence | Reverse sequence | AT |
|------|-----------|------------------|------------------|-----|
| I | Xpm 2-22F | CAATGGACCGCCTCGCTGGC | VIC-GTGGGTGCCTGGTCGCAACG | 63 |
| | Xpm 2-18F | TTTCGGGGTGGCGGCATTGT | 6-FAM-GCGATTGATCGCCGGGTCGT | 63 |
| | Xpm 2-35F | TGACCCTGAAGGGCGAGGGC | PET-CGCCGGCAGCAACTGTCCAC | 63 |
| II | Xpm 2-5F | AACGCAGCGGGTGTGGTTGC | PET-GCAAGCAGCGCAGAAGGCGA | 64 |
| | Xpm 2-33F | GCGGCTTCGTCAGTACCCTC | VIC-CGCAATGCTCAAATCGCCCT | 64 |
| | Xpm 2-29F | GTCGGCCTGTGGTGGCGGAG | 6-FAM-TTTCCGGCAACTGGCAGGCG | 64 |
| III | Xpm 2-3F | ACCGTGCCCATTCCGGCACC | PET-CCATTACCACCGCTGCGGGC | 62 |
| | Xpm 2-20F | CGGTATGGGGCCCCGAAAGC | VIC-CTCCGTGCACACCCGGCACT | 62 |
| | Xpm 2-23F | CGACGTGCGTGCGTAGGCGA | 6-FAM-GCGTGAGGCAAACATCGGCG | 62 |

PET, and VIC, Applied Biosystems) (S2 Table) and detected by capillary electrophoresis in an ABI 3500 Genetic Analyzer as indicated above.

## Data scoring

Fragment sizes for each VNTR locus were estimated using GENEMAPER v 4.0 (Applied Biosystems). When necessary, a truncated number of repeats was rounded up to the nearest integer [12]. The discriminatory power of each VNTR locus was calculated based on the Hunter and Gaston discriminatory index (HGDI) [42]. Numbers of alleles and Nei's unbiased estimates of diversity were estimated using ARLEQUIN software [43]. Haplotype networks were represented by minimum spanning trees using the algorithm combining global optimal eBURST (goeBURST) and Euclidian distances in the software PHYLOViZ 2.0 [44]. The minimum spanning trees display the relationships between clonal complexes defined as groups of single locus variants (SLVs). The population structure of *Xpm* was assessed by discriminant analysis of principal components (DAPC) of MLVA-12 data using the adegenet V.2.1.1. R package [45,46]. DAPC is free of any assumption linked to a population genetic model (e.g., Hardy-Weinberg equilibrium or linkage equilibrium), and, thus, is suited for analysis of datasets produced from predominantly clonal bacteria.

## Pathogenicity tests

Pathogenicity tests were performed with strains representative of different haplotypes and clonal complexes selected from the minimum spanning tree based on the MLVA-8 minisatellites scheme. The strains selected were CIAT1205, CIAT1135, UA1591, CIAT1241, UA556, UA2146, CIAT1202 and CIO151. Two month-old plants of the cassava varieties CM523-7, COL1505, NGA11, CM6438-14 and cv.60444 were inoculated. Ten μl of *Xpm* cell suspension (optical density at 600nm ($OD_{600nm}$) = 0,02, ~$10^7$ colony forming units (CFU ml$^{-1}$) were inoculated in the stem. A scale of symptoms from 0 to 5 (0 = no symptoms, 1 = necrosis at the inoculation point, 2 = stem exudates, 3 = one or two wilted leaves, 4 = more than three wilted leaves, and 5 = plant death) was used. Symptom development was monitored at 7, 14, 21, and 28 days after inoculation. Disease progression AUDPC (area under disease progress curve) [47] was calculated for each of the five replicates. The determination of the resistance level of the cassava varieties was based on four categories, as defined previously [36,48]. A variety was considered resistant when $\Sigma$AUPDC $\leq$ 39, moderately resistant: 39 < $\Sigma$ AUDPC $\leq$ 44), moderately susceptible: 44 < $\Sigma$AUDPC < 49 and susceptible when $\Sigma$AUDPC $\geq$ 49.

# Results

## Selection of eight new minisatellite loci

After the screening of 12 *Xpm* genomes, [49] VNTR loci met the criteria of the search. Of these candidates, only ten loci were selected because their flanking regions were almost fully conserved, and their sequences were identical for each locus in all the strains analyzed. Five TR loci did not show any polymorphism for these 12 genomes. For the polymorphic loci the HGDI was relatively high varying between 0.167 and 0.849 with an allele number ranging from two to six, respectively (Table 2).

The total length of most of the loci including the primer regions ranged between 197 to 490 bp. The unit length of the tandem repeat ranged between 10 and 26 bp. Individual bands were observed for each primer and in PCR multiplex (Fig 1). However, for loci Xpm2-29 and Xpm2-2, the size of amplicons exceeded the range size covered by capillary electrophoresis. The sequencing analysis of these two loci showed large insertions so they were removed from the typing scheme. The final minisatellite scheme was formed by eight VNTR loci and called hereafter MLVA-8 (Table 1). This scheme had three perfect loci (Xpm 2–20, Xpm 2–23, and Xpm 2–33) and five imperfect loci (Table 2).

## MLVA-8 genotyping of a world collection of *Xpm* strains

Thirty-one multilocus haplotypes were obtained from the worldwide collection of 93 *Xpm* strains using the MLVA-8 scheme. All the TR loci were polymorphic except Xpm2-5, which we hypothesize was fixated by natural selection. Polymorphic loci produced between two (loci Xpm2-33 and 2–35) and eight alleles (Xpm2-18) with Nei's gene diversity varying from 0.108 to 0.831 (S3 Table). TR markers along the evolutionary path of the minimum spanning tree were analyzed by assessing the number of repeats involved in the polymorphism of recently diverging haplotypes, i.e. SLVs. Multiple-repeat variations occurred only for loci Xpm 2–3 and

**Table 2. Characteristics of the minisatellite loci selected in this study from 12 genomes of *Xanthomonas phaseoli* pv. *manihotis*.**

| VNTR Locus | Motif sequence | FS[a] | Unit length (bp) | # Alleles | Allelic range | HGDI[b] | Other, previously used nomenclature[c] of VNTR locus |
|---|---|---|---|---|---|---|---|
| **Xpm 2–2** | GGTTCTAGCTTTTA, GGTTCGTGCTTCTA, GGTTCTTGCTTCTC | 9 | 14 | 3 | 4–6 | 0.639 | |
| **Xpm 2–3** | CGGCCCCGGACGT(1), CGGCCCCGCGTCC(1) | 10 | 12 | 1 | 4 | 0 | |
| **Xpm 2–5** | GTTGGCCGGTTCGGTTA(1) GTTGGCCGGTTCGGTAG(1) | 12 | 17 | 1 | 3 | 0 | |
| **Xpm 2–18** | CACCGCCACTACG(1) CACCACCACAACG(1) CACCGCCACAACG (9) | 12 | 13 | 6 | 5–11 | 0.849 | XaG2_52 |
| **Xpm 2–20** | CAACATCCACAG(2) | 12 | 12 | 2 | 3–5 | 0.167 | Xpm 1–20 |
| **Xpm 2–22** | GCAAGCGCGGTGCAACCACGTA(1) GCAAGCGCAGTGCAACCACGCG(1) GCAAGCGCAGTGCAGCGACGCA(1) | 12 | 22 | 2 | 3–4 | 0.303 | |
| **Xpm 2–23** | AGATCGAGACACGC(2) | 12 | 14 | 4 | 4–7 | 0.636 | |
| **Xpm 2–29** | CGCCGGCACCTG(2) | 10 | 12 | 1 | 6 | 0 | |
| **Xpm 2–33** | GAGGCTTGCGTGTCCTTGCGTGTGAG (2) | 9 | 26 | 1 | 3 | 0 | XaG2_109 |
| **Xpm 2–35** | GCTGCCGGCA(1) GCCGCCTGCA(1) GGTGCCGCAC(1) CTGCAGGCGG(1) CTGCCGGCAG (1) | 10 | 10 | 1 | 4 | 0 | Xpm 1–35 |

[a] FS: Flanking sequences (number of genomes out of the 12 for whose flanking sequences were detected).

[b] HGDI: Hunter-Gaston discriminatory index.

[c] Nomenclature from Trujillo et al. [35] and/or from Arrieta-Ortiz et al. [41] and/or Rache et al. [40].

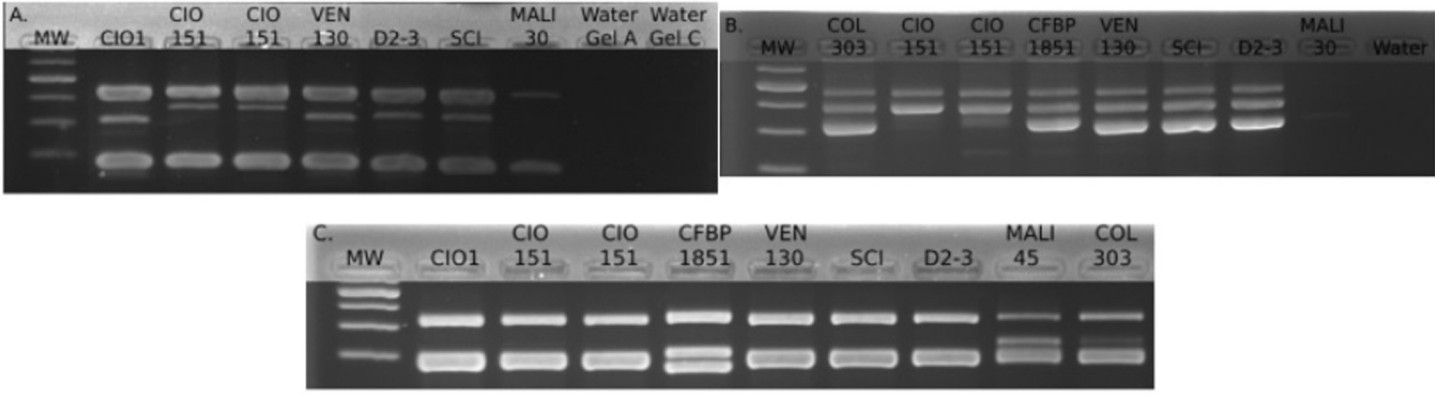

**Fig 1. Visualization of multiplex PCR amplicons on agarose gels.** MW: Molecular marker. A. Pool I, loci Xpm 2–22, Xpm 2–18 and Xpm 2–35. B. Pool II, loci Xpm 2–33, Xpm 2–5 and Xpm 2–29. C. Pool III loci Xpm 2–3, Xpm 2–20 and Xpm 2–23.

Xpm 2–20 with 50% of double- or triple-repeat variations and 33% of double repeat variations, respectively. Changes consisting of a single-repeat variation represented 85.2% of the SLVs. Our data suggest that these TR minisatellites mainly evolve following a generalized stepwise mutation model (or two-phase model) [48].

Eighteen haplotypes out of 31 originated from a unique country. Thirteen haplotypes were shared by strains originating from different countries, six of which were from South American countries and seven from different continents. All the African strains shared haplotypes with South American strains except MLVA-8 haplotype #5 composed only of Burkinabe strains. The minimum spanning tree separated the haplotypes in three clonal complexes (CC, *i.e.* clusters of single locus variants), and one singleton, i.e. haplotypes with no SLVs (Fig 2). The main MLVA-8 CC1 grouped 24 haplotypes including all the African strains and strains from the other countries included in the analysis, except a single strain from China. They were isolated from 1966 to 2016. Three MLVA-8 haplotypes were recovered over a period of more than 30 years (# 1, 7 and 16) (S1 Table and Fig 2). All haplotypes originating from Mali or from Burkina-Faso were SLVs (MLVA haplotypes #2 and 4 differ only by Xpm 2–18). The MLVA-8 CC2 (n = 4 haplotypes), a triple locus variant of CC1, was composed by the Chinese strain and strains from Brazil and New Zealand that were all isolated between 1974 and 1996. The third CC (2 haplotypes) and the singleton, haplotype #21, grouped strains from Colombia and New Zealand isolated between 1966 and 1974. Strains from China and Vietnam did not share any haplotypes with other strains.

## MLVA-14 genotyping of a world collection of *Xpm* strains

We compared the discriminatory power of MLVA-8 with the previously reported MLVA-14 microsatellite scheme. MLVA-14 scheme was able to distinguish 89 haplotypes out of the 93 *Xpm* strains. All loci were highly polymorphic with a mean allelic richness of A = 13.5 leading to high discriminatory index (HGDI = 0.999) and Nei's gene diversity index ($H_E$ = 0.863), which were much higher than those estimated from the MLVA-8 dataset (Tables 3 and S4). Strains sharing the same haplotype originated from the same country and, when the information was available, they were isolated the same year in the same locality (S1 Table).

The minimum spanning tree describing the relationships between the 89 haplotypes produced ten small clonal complexes grouping only two or three SLVs (S1 Fig). All these CCs grouped epidemiologically related strains from the same region isolated the same year except

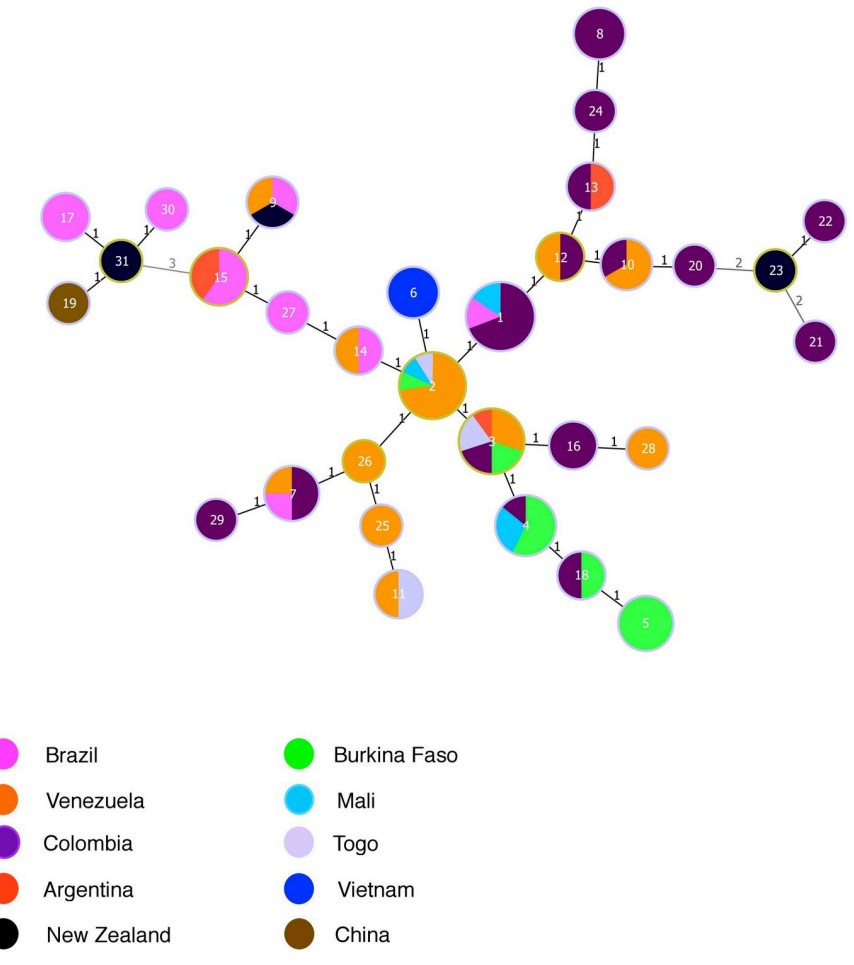

**Fig 2. Minimum spanning tree displaying relationships between haplotypes using the MLVA-8 minisatellite scheme.** Colors indicate the geographical origin of the haplotype, and the circle size indicates the number of strains of each haplotype. Numbers indicate the number of locus variants between haplotypes.

for the two MLVA-14 haplotypes #3 and 5 (CC7) from two different Colombian regions and isolated during two consecutive years (S1 Table). Other closely related haplotypes (*i.e.* DLVs) also originated from the same country with the exception of two strains from New Zealand isolated in 1966 (haplotypes #44 and 69) which are DLVs of strains from Venezuela and Colombia isolated in the beginning of the seventies, respectively (S2 Fig and S1 Table). Almost all the other haplotypes differed from each other by at least four TR loci but mostly gathered by

**Table 3. Estimators of diversity for each MLVA scheme from the *Xanthomonas phaseoli* pv. *manihotis* world collection (n = 93).**

| Scheme | Polymorphic loci | MLGs[a] | Mean allelic richness A | HGDI[b] | $H_T$ [c] | Simpson' index |
|---|---|---|---|---|---|---|
| **MLVA-8** | 7/8 | 31 | 3.375 | 0.944 | 0.335 | 0.937 |
| **MLVA-12** | 11/12 | 78 | 4.750 | 0.995 | 0.502 | 0.985 |
| **MLVA-14** | 14/14 | 89 | 13.5 | 0.999 | 0.867 | 0.988 |

[a] MLGs, number of multilocus genotypes.

[b] HGDI Hunter and Gaston discriminatory index.

[c] $H_T$ Nei's total gene diversity.

country or by continent. The closest African and South American strains, from Venezuela or Colombia, were separated by variations at nine TR loci.

## A combined MLVA-12 scheme proposed for large scale epidemiology

Four VNTRs (2–31,2–6, 2–7, 2–38) among the least discriminant of the MLVA-14 scheme were combined to the eight VNTRs (2–22, 2–18, 2–35, 2–20, 2–3, 2–23, 2–5, 2–33) of the MLVA-8 scheme to slightly increase its discriminatory power. The mean allelic richness and HGDI produced by the MLVA-12 dataset were estimated to 4.75 and 0.995, respectively (Table 3). Seventy-eight MLVA-12 haplotypes out of 93 strains were separated into 16 CCs (n = 60 strains) composed of two to seven haplotypes and 26 singletons (n = 33 strains) (S2 Fig). Nine CCs contained strains originating from the same country among which six consisted of strains isolated the same year in the same region probably corresponding to the same outbreak. Five CCs grouped strains from the same continent either South America (CCs 6, 11 and 16, each with strains from two different countries) or Africa (CCs 1 and 2, both with strains originated from Mali, Burkina Faso and Togo). Most of these CCs are tightly related as they differ by variations in two loci. We used DAPC to delineate seven clusters in our dataset (Fig 3). The assignation probability to clusters was > 0.99 for all strains in the dataset. Interestingly, the large polymorphism revealed for strains from South America split them in the seven clusters. Strains originating from Argentina (n = 4), Brazil (n = 12), Colombia (n = 28) and Venezuela (n = 21) were assigned to three (Argentina) or four (other countries) distinct clusters. Cluster #1 included strains from South America (Argentina, Brazil and Colombia) together with strains from Viet Nam. Cluster #2 mostly included strains from South America (Argentina, Brazil and Venezuela) together with a single strain from New Zealand. All African strains gathered in cluster #3 together with some strains from Colombia and Venezuela. Clusters #4 and 5 solely included strains from Colombia and Venezuela, respectively. Cluster #6 primarily included strains from South America (Brazil and Colombia) together with a few strains from China and New Zealand. Cluster #7 included strains from all countries in South America (Argentina, Brazil, Colombia and Venezuela). The structure revealed by DAPC was overall consistent with the produced MST (Figs 3 and S2).

## Pathogenicity patterns are not associated to genetic groups

The AUDPC values were used to determine if cassava varieties are resistant ($\Sigma$AUPDC $\leq$39), moderately resistant (39<$\Sigma$AUDPC<44), moderately susceptible (44<$\Sigma$AUDPC<49), or susceptible ($\Sigma$AUDPC$\geq$49) to the eight *Xpm* strains evaluated (Table 4 and S3 Fig). One cultivar, cv. 60444, was susceptible to all strains tested and two varieties (NGA11 and CM523-7) were susceptible to nearly all strains tested. On the other hand, cultivars COL1505 and CM6438-14 were resistant or moderately resistant to at least three of the strains evaluated (Table 4). Regarding the strain pathogenicity level, strains CIAT1202, UA556, UA2164, CIO151 and CIAT1241 caused disease in all inoculated varieties. On the opposite hand, strain CIAT1135 was poorly pathogenic. Based on this information, pathological patterns were classified as follows. The more virulent strains (CIAT1202, UA556, UA2164, CIO151 and CIAT1241) were classified in group I. Strains with intermediate virulence (UA1591 and CIAT1205) were designated in group II. The least virulent strain CIAT1135 was assigned to group III (Table 4).

Data shown is the mean of the sum of AUDPC in arbitrary units at 28 dpi from five replicates per genotype. A genotype was considered resistant (R) if $\Sigma$AUPDC $\leq$39, moderately resistant (MR) if 39<$\Sigma$AUDPC<44, moderately susceptible (MS) when 44<$\Sigma$AUDPC<49 and susceptible (S) when $\Sigma$AUDPC$\geq$49, NE: Not evaluated.

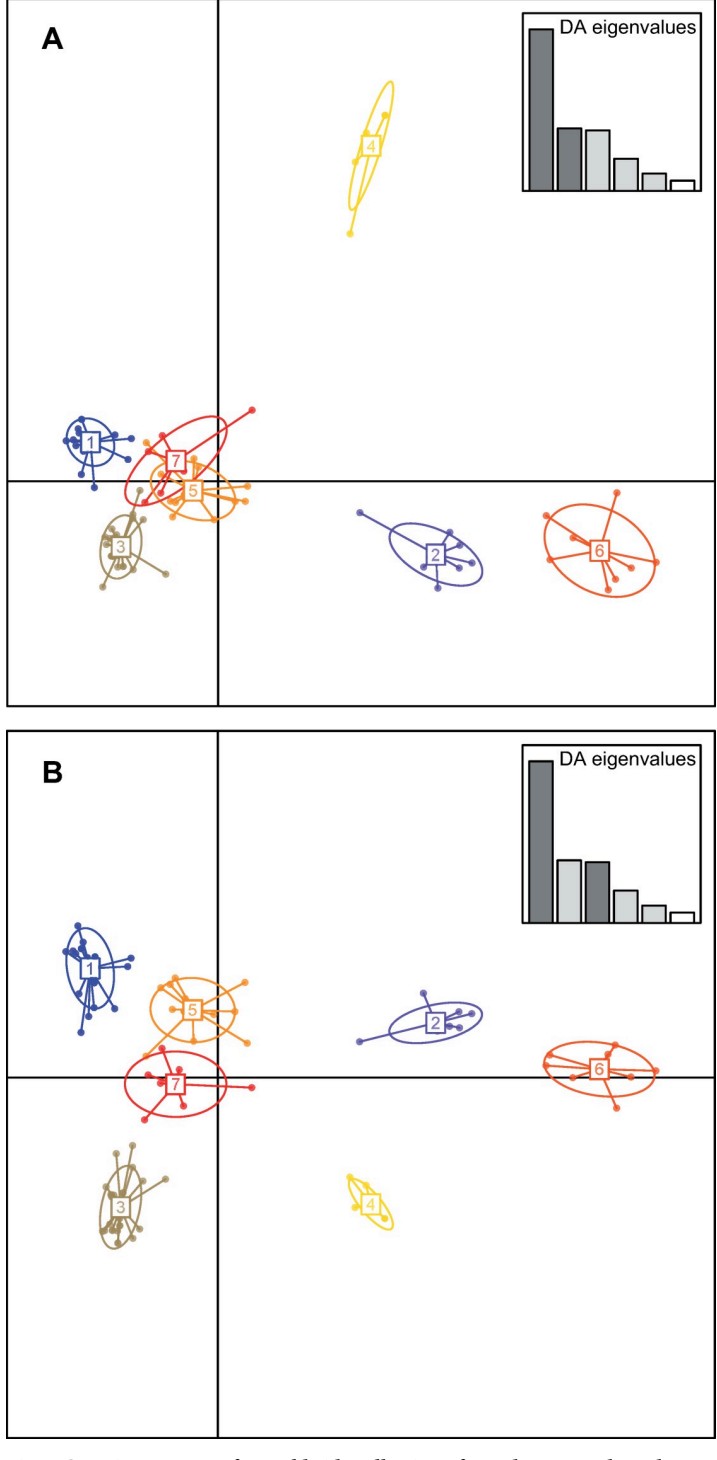

**Fig 3. Genetic structure of a worldwide collection of *Xanthomonas phaseoli* pv. *manihotis* based on the discriminant analysis of principal components (DAPC) of minisatellite and microsatellite data (MLVA-12).** Numbers and colors represent the seven genetic clusters retained from Bayesian information criterion (BIC) values. (A) Scatterplot representing axes 1 and 2 of the DAPC. (B) Scatterplot representing axes 1 and 3 of the DAPC.

**Table 4. Pathogenicity tests performed using eight representative *Xpm* strains.**

| Strain | Cassava Variety | | | | | Relative virulence group | Haplotype MLVA-8 (MLVA-12) | Country |
|---|---|---|---|---|---|---|---|---|
| | CM523-7 | NGA11 | COL1505 | CM6438-14 | cv.60444 | | | |
| CIAT1202 | S (58,5±3,4) | S (66,7±1,1) | S (61,8±5,0) | NE | S (52±3,1) | I | 13 (30) | Colombia |
| UA556 | S (68,2±7,6) | S (56,4±5,1) | MS (47,2 ±11,2) | S (69,1±7,3) | S (73,3±3,8) | I | 4 (11) | Colombia |
| UA1591 | R (29,9±8,9) | S (49,6±6,3) | MR (41,9±8,8) | MR (42,4 ±2,4) | NE | II | 7 (59) | Colombia |
| UA2164 | S (55,6±8,5) | S (81±2,5) | S (60,9±4,5) | NE | NE | I | 1 (7) | Colombia |
| CIO151 | S (59,3±1,2) | S (69,1 ±10,1) | S (53,1±5,5) | S (50,7±2,0) | MS (44,9 ±6,5) | I | 18 (18) | Colombia |
| CIAT1135 | R (5,2±2,0) | R (20,3±3,7) | R (0) | R (22,9±2,4) | NE. | III | 19 (16) | China |
| CIAT1205 | MS (47,1 ±3,0) | S (67,2±3,1) | R (26,3±6,2) | NE | MS (48,1 ±8,9) | II | 23 (25) | New Zealand |
| CIAT1241 | S (67±5,8) | S (68,5±2,5) | S (53,5±6,0) | S (55,4±7,4) | NE | I | 15 (58) | Argentina |

## Discussion

Epidemiology can be deciphered at different time and space scales by selecting the level of genomic resolution of the markers to monitor either the geographic spread of inoculum or the prevalence of certain epidemic or endemic clones for surveillance [2–4,6]. Cassava Bacterial Blight is a destructive disease widely distributed in the different tropical areas where cassava is grown. It has recently been reported in new areas [50]. A global molecular epidemiology analysis is needed to better understand the expansion of the disease for improving the monitoring of CBB. We recently developed a MLVA scheme targeting microsatellites with a high discriminatory power expected to be more adapted to local epidemiology and outbreak analysis [40]. Here, we evaluated new minisatellites for their usefulness in describing the relationships of non-epidemiologically related strains and large-scale epidemiology. These tandem repeat sequences, i.e. micro- and minisatellites, show motifs whose length variability is one of the main factors known to influence the mutation rate of VNTRs [14]. We propose a MLVA-12 scheme which takes advantages of combining TR loci with different mutation rates for global epidemiological surveillance.

Eight minisatellites (MLVA-8) with a unit length ranging from 10 to 26 bp (i.e. markers with an expected low mutation rate) separated a world collection of *Xpm* strains isolated over a period of fifty years (1966–2016) in three CCs and a singleton. The major CC1 grouped strains originating from all the represented countries, except the unique Chinese strain that was included in our collection. All the strains from Africa grouped in this CC while strains from South America, the continent where the disease probably originated, were distributed in the three CCs. The MLVA-8 dataset confirmed (i) a more extensive genetic diversity for the South American strains than for the African strains and (ii) the hypothesis of some genetic links between African strains and some South-American strains as previously shown using four RFLP probes and SNPs based-phylogeny [26,37].

The MLVA-8 scheme alone lacked discriminatory power and therefore did not reveal a fully meaningful epidemiological perspective at a global scale. The relatively low number of minisatellite loci identified and their low mutation rates both are factors favoring homoplasy [51,52]. In contrast, 89 out of 93 *Xpm* strains genotyped using the MLVA-14 scheme were separated as unique haplotypes clearly confirming the high mutation rate of this genotyping technique, consistent with a previous study and suitable for small scale epidemiology [40].

A MLVA-12 scheme combining the eight minisatellites and four microsatellites from the MLVA-14 and displaying the lowest evolution rates is proposed to increase discriminatory power and minimize homoplasy effects [51,52]. In consequence the MLVA-12 scheme, will increase the genetic space of possible patterns as compared to MLVA-8 and would be suitable for epidemiological surveillance. The variable mutation rates within the MLVA-12 scheme allowed a nested approach where the genetic relationships among strains is described within deeper phylogenetic clusters. Such approaches, when combining markers with different discriminatory power, progressively or simultaneously, have been found to increase the accuracy of the phylogeny and to minimize the homoplasy effects [4,52,53].

A DAPC analysis of MLVA-12 data separated the haplotypes in seven distinct clusters. Interestingly, this analysis placed strains from South America in all clusters. This further supports the wide genetic diversity already described in South America (*i.e.* center of origin of cassava and the continent where CBB was firstly reported [26,30,33,37]. In contrast, all African strains studied here grouped in a single cluster also containing strains from South America, suggesting a possible relatedness and the existence of a common ancestor. However, based on MLVA-14, the amount of genetic relatedness between African and the closest South American strains did not support recent epidemiological links. Cassava was introduced in Africa during the seventeenth and eighteenth centuries at a period where regular communications already existed between the two continents [54]. After its first description in Brazil in 1912, cassava bacterial blight was first reported in continental Africa in the early 1970's [24,54]. The pathogenic bacterium was likely introduced in this continent through transportation of infected plant material after multiple introduction events into both Eastern and Western African coasts. Three strains isolated in New-Zealand during outbreaks in 1960s or in 1980 were distributed in two different DAPC clusters, one of them sharing an MLVA-12 haplotype with a strain from Venezuela and the others were SLVs of Colombian and Brazilian strains. This result supports the occurrence of multiple introduction events of *Xpm* in this country.

No clear correlation between the MLVA profiles and virulence patterns occurred. The different haplotypes representative of the genetic variability of observed within our world *Xpm* collection from MLV8-dataset produced different pathological patterns though all the isolates were virulent on the Colombian cassava varieties. Such variability in the pathogenicity of *Xpm* from an international collection without any link between the genetic and pathological patterns was previously reported using different molecular markers [55]. Variations of strain aggressiveness and a lack of correlation between the genetic and pathological patterns were further described at a regional or local scale in different countries revealing dynamic spatio-temporal changes within the *Xpm* populations [29,30,33,36,38].

## Conclusions

The MLVA-12 scheme proposed in this study combined both minisatellite and microsatellite loci that brought complementary information based on their different discriminatory power. *Xpm* is a monomorphic plant pathogen widely spread in tropical regions. This portable and resolving genotyping tool could be useful to further investigate the global epidemiology of this plant pathogenic bacterium of cassava, whose improved understanding will require additional data.

## Supporting information

**S1 Fig. Minimum spanning tree displaying the relationships between haplotypes using the MLVA-14 Microsatellite scheme.** Colors indicate the origin of the haplotype, and the circle size indicates the number of strains of each haplotype. Blue circles indicate clonal complexes

and orange circles indicate groups of double locus variants. Numbers indicate the number of loci variants between haplotypes.
(TIF)

**S2 Fig. Minimum spanning tree displaying the relationships between haplotypes using the MLVA-12 scheme.** Colors indicate the origin of the haplotype, and the circle size is relative to the number of strains of each haplotype. The solid lines represent clonal complexes (CCs) grouping single locus variants and dotted lines group up to double locus variants. Numbers indicate the number of loci variants between haplotypes.
(TIF)

**S3 Fig. Disease symptoms for resistant (R), moderately resistant (MR), susceptible (S), and moderately susceptible (MS) cultivars.**
(TIF)

**S4 Fig. Original images of agarose gel from Fig 1.**
(TIF)

**S1 Table. Worldwide collection of *Xanthomonas phaseoli* pv. *manihotis* strains analyzed in this study and haplotypes obtained from the different MLVA schemes.**
(DOCX)

**S2 Table. Multiplex scheme and primer pairs used in the MLVA-14 scheme targeting microsatellites.**
(DOCX)

**S3 Table. Diversity indices for MLVA-8 scheme from genotyping of a worldwide collection of *Xpm* (n = 93).**
(DOCX)

**S4 Table. Diversity indices for MLVA-14 scheme from genotyping of a worldwide collection of *Xpm* (n = 93).**
(DOCX)

## Acknowledgments

*In memoriam* of Christian Vernière, who was a wonderful person and researcher.

## Author Contributions

**Conceptualization:** Ralf Koebnik, Adriana Bernal.

**Data curation:** Laurence Blondin, Paula Diaz Tatis, Andrea Camargo.

**Formal analysis:** Leidy Rache, Laurence Blondin, Paula Diaz Tatis, Andrea Camargo, Camilo López, Olivier Pruvost, Christian Vernière.

**Funding acquisition:** Boris Szurek, Ralf Koebnik, Silvia Restrepo, Adriana Bernal.

**Investigation:** Leidy Rache.

**Methodology:** Leidy Rache, Paula Diaz Tatis, Carolina Flores, Andrea Camargo, Moussa Kante, Issa Wonni, Camilo López, Ralf Koebnik, Adriana Bernal.

**Project administration:** Adriana Bernal.

**Supervision:** Ralf Koebnik, Silvia Restrepo, Adriana Bernal.

**Validation:** Ralf Koebnik.

**Writing – original draft:** Leidy Rache, Paula Diaz Tatis, Camilo López, Olivier Pruvost, Ralf Koebnik, Silvia Restrepo, Adriana Bernal, Christian Vernière.

**Writing – review & editing:** Leidy Rache, Boris Szurek, Stephane Dupas, Olivier Pruvost, Ralf Koebnik, Silvia Restrepo, Adriana Bernal, Christian Vernière.

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
