## [Decision Letter · Decision Letter 0]

17 Jan 2023

PONE-D-22-33979A minisatellite-based MLVA for deciphering the global epidemiology of the bacterial cassava pathogen Xanthomonas phaseoli pv. manihotisPLOS ONE

Dear Dr. Bernal,

Thank you for submitting your manuscript to PLOS ONE. After careful consideration, we feel that it has merit but does not fully meet PLOS ONE’s publication criteria as it currently stands. Therefore, we invite you to submit a revised version of the manuscript that addresses the points raised during the review process.

We look forward to receiving your revised manuscript.

Kind regards,

Tushar Shaw

Academic Editor

PLOS ONE

Journal Requirements:

2. We note that this study relies on the analysis of worldwide collection of 93 Xpm strains isolated over a period of fifty years. For reproducibility purposes, please provide the references in Supp Table 1 of all the isolates (if already published) and/or please provide further information about how others may be able to access the strains. If available on a database, please specify which one.

"We thank the Faculty of Sciences from Universidad de los Andes-Bogotá, Colombia (INV-2021-128-2283), the UMR Interactions Plantes Microorganismes Environnement, and the Agropolis Foundation (project PAIX, grant no. 1 403-073), Montpellier, France, for financial support. The Ecos Nord programme supported inter-laboratory mobility (grant no. C15A01). Leidy Rache was supported by Colciencias with a doctoral fellowship (call No. 528, 2011). Olivier Pruvost acknowledges the European Union (ERDF contract GURDT I2016‐1731‐0006632) and Réunion regional council for support."

"We thank the Faculty of Sciences from Universidad de los Andes-Bogotá, Colombia (INV-2021-128-2283), the UMR Interactions Plantes Microorganismes Environnement, and the Agropolis Foundation (project PAIX, grant no. 1 403-073), Montpellier, France, for financial support. The Ecos Nord programme supported inter-laboratory mobility (grant no. C15A01). Leidy Rache was supported by Colciencias with a doctoral fellowship (call No. 528, 2011). Olivier Pruvost acknowledges the European Union (ERDF contract GURDT I2016‐1731‐0006632) and Réunion regional council for support."

"We thank the Faculty of Sciences from Universidad de los Andes-Bogotá, Colombia (INV-2021-128-2283), the UMR Interactions Plantes Microorganismes Environnement, and the Agropolis Foundation (project PAIX, grant no. 1 403-073), Montpellier, France, for financial support. The Ecos Nord programme supported inter-laboratory mobility (grant no. C15A01). Leidy Rache was supported by Colciencias with a doctoral fellowship (call No. 528, 2011). Olivier Pruvost acknowledges the European Union (ERDF contract GURDT I2016‐1731‐0006632) and Réunion regional council for support."

7. We note that you have included the phrase “data not shown” in your manuscript. Unfortunately, this does not meet our data sharing requirements. PLOS does not permit references to inaccessible data. We require that authors provide all relevant data within the paper, Supporting Information files, or in an acceptable, public repository. Please add a citation to support this phrase or upload the data that corresponds with these findings to a stable repository (such as Figshare or Dryad) and provide and URLs, DOIs, or accession numbers that may be used to access these data. Or, if the data are not a core part of the research being presented in your study, we ask that you remove the phrase that refers to these data.

8. We note that you have referenced (ie. Rache L, Blondin L, Flores C, Trujillo C, Szurek B, Restrepo S, et al. [40]) which has currently not yet been accepted for publication. Please remove this from your References and amend this to state in the body of your manuscript: (ie “Rache L. et al. [Unpublished]”) as detailed online in our guide for authors

http://journals.plos.org/plosone/s/submission-guidelines#loc-reference-style "

9. Please upload a new copy of Figure 3 as the detail is not clear. Please follow the link for more information:

https://blogs.plos.org/plos/2019/06/looking-good-tips-for-creating-your-plos-figures-graphics/

https://blogs.plos.org/plos/2019/06/looking-good-tips-for-creating-your-plos-figures-graphics/

**Additional Editor Comments:**

The authors have aimed to identify versatility of TR markers in order to develop improved MLVA schemes with desirable characteristics for outbreak investigation or epidemiological surveillance of Xpm. The study has been well planned but needs some minor revisions before it can be accepted for publicaition.

The authors need to address the comments of the reviewers as mentioned below:

Reviewer 1:

The authors developed a MLVA-12 scheme combining both minisatellite and microsatellite loci with different discriminatory power. This scheme is useful for studying the genetic diversity of Xanthomonas phaseoli

pv. manihotis in the world. Statistical analyses were correctly performed. The authors also showed that the pathogenicity of the strains is not related to their genetic profiles.

Apart from several errors in the article highlighted below, the work is of good quality.

line 171 : « 3 loci in each multiplex » but in table 1 : only 2 loci are present in Pool II. So one loci is missing in table 1, Pool II. This pool is correct in Figure 2.

Line 168-170 : primers were tested with DNA from 9 strains (4 from Columbia, 1 from Venezuela, 1 from Ivory Coast, 1 from Burkina, 2 from Mali). Lines 170-170 : these strains are not all in the S1 table. Why ? if they have been tested…

Line 195 : Xpm 2-20 is not in [40], it’s Xpm 1-20. Modify sentence

Line 237 : « 11 loci were selected » but only 10 loci are in Table 2. Modify sentence

Line 253-254 : « for loci Xpm1-29 and Xpm1-2 » : is it not rather Xpm2-29 and Xpm2-2 (from Table 2) ?

Line 262-265 : it seem's that the names of the loci are wrong, please replace 1 with 2.

Line 311 : « Four VNTRs among the least discriminant of the MLVA-14 scheme were combined » : please name them in the sentence.

Table 2 : please explain what means the number between brackets after the motif sequence and why some letters are in bold for Xpm 2-2

Reviewer 2:

The manuscript “A minisatellite-based MLVA for deciphering the global epidemiology of the bacterial cassava pathogen Xanthomonas phaseoli pv. manihotis” describes a novel MLVA scheme for epidemiology of the cassava pathogen. At the same time, it provides a novel insight into the global population structure of the pathogen in an intra-pathovar level. The study is well designed, experimental procedure is fine, and the text reads smoothly. The authors could add a figure including disease symptoms on resistant, moderately resistant, and moderately susceptible cultivars.

Reviewers' comments:

Reviewer's Responses to Questions

**Comments to the Author**

1. Is the manuscript technically sound, and do the data support the conclusions?

Reviewer #1: Yes

Reviewer #2: Yes

2. Has the statistical analysis been performed appropriately and rigorously? 

Reviewer #1: Yes

Reviewer #2: Yes

3. Have the authors made all data underlying the findings in their manuscript fully available?

Reviewer #1: Yes

Reviewer #2: Yes

4. Is the manuscript presented in an intelligible fashion and written in standard English?

Reviewer #1: Yes

Reviewer #2: Yes

5. Review Comments to the Author

Reviewer #1: The authors developed a MLVA-12 scheme combining both minisatellite and microsatellite loci with different discriminatory power. This scheme is useful for studying the genetic diversity of Xanthomonas phaseoli

pv. manihotis in the world. Statistical analyses were correctly performed. The authors also showed that the pathogenicity of the strains is not related to their genetic profiles.

Apart from several errors in the article highlighted below, the work is of good quality.

line 171 : « 3 loci in each multiplex » but in table 1 : only 2 loci are present in Pool II. So one loci is missing in table 1, Pool II. This pool is correct in Figure 2.

Line 168-170 : primers were tested with DNA from 9 strains (4 from Columbia, 1 from Venezuela, 1 from Ivory Coast, 1 from Burkina, 2 from Mali). Lines 170-170 : these strains are not all in the S1 table. Why ? if they have been tested…

Line 195 : Xpm 2-20 is not in [40], it’s Xpm 1-20. Modify sentence

Line 237 : « 11 loci were selected » but only 10 loci are in Table 2. Modify sentence

Line 253-254 : « for loci Xpm1-29 and Xpm1-2 » : is it not rather Xpm2-29 and Xpm2-2 (from Table 2) ?

Line 262-265 : it seem's that the names of the loci are wrong, please replace 1 with 2.

Line 311 : « Four VNTRs among the least discriminant of the MLVA-14 scheme were combined » : please name them in the sentence.

Table 2 : please explain what means the number between brackets after the motif sequence and why some letters are in bold for Xpm 2-2

Reviewer #2: The manuscript “A minisatellite-based MLVA for deciphering the global epidemiology of the bacterial cassava pathogen Xanthomonas phaseoli pv. manihotis” describes a novel MLVA scheme for epidemiology of the cassava pathogen. At the same time, it provides a novel insight into the global population structure of the pathogen in an intra-pathovar level. The study is well designed, experimental procedure is fine, and the text reads smoothly. The authors could add a figure including disease symptoms on resistant, moderately resistant, and moderately susceptible cultivars.

6. PLOS authors have the option to publish the peer review history of their article (what does this mean?). If published, this will include your full peer review and any attached files.

Reviewer #1: No

Reviewer #2: **Yes: **Ebrahim Osdaghi

---

## [Author Response · Author response to Decision Letter 0]

10 Mar 2023

March 8th, 2023

Tushar Shaw

Academic Editor

PLOS ONE

We received the reviewer´s comments on the manuscript entitled ‘A minisatellite-based MLVA for deciphering the global epidemiology of the bacterial cassava pathogen Xanthomonas phaseoli pv. manihotis’. We truly appreciate the comments, and we found them all very helpful. 

We are sending a revised version of the paper where changes have been done considering their comments. Modifications in the manuscript were included with track changes mode. The response to the reviewers’ comments appear below.

Response to reviewer(s)' Comments:

Journal Requirements:

Response: Thanks for recommendations, the manuscript was adjusted according to PLOS ONE´s style requirements.

2. We note that this study relies on the analysis of worldwide collection of 93 Xpm strains isolated over a period of fifty years. For reproducibility purposes, please provide the references in Supp Table 1 of all the isolates (if already published) and/or please provide further information about how others may be able to access the strains. If available on a database, please specify which one.

Response: We included this sentence: “The strains are available upon request to B. Szurek at IRD”.

Response: Thank you, the corrected information was included in the resubmitted version. 

"We thank the Faculty of Sciences from Universidad de los Andes-Bogotá, Colombia (INV-2021-128-2283), the UMR Interactions Plantes Microorganismes Environnement, and the Agropolis Foundation (project PAIX, grant no. 1 403-073), Montpellier, France, for financial support. The Ecos Nord programme supported inter-laboratory mobility (grant no. C15A01). Leidy Rache was supported by Colciencias with a doctoral fellowship (call No. 528, 2011). Olivier Pruvost acknowledges the European Union (ERDF contract GURDT I2016‐1731‐0006632) and Réunion regional council for support."

Response: the sentence “The funders had no role in study design, data collection and analysis, decision to publish, or preparation of the manuscript" was included.

"We thank the Faculty of Sciences from Universidad de los Andes-Bogotá, Colombia (INV-2021-128-2283), the UMR Interactions Plantes Microorganismes Environnement, and the Agropolis Foundation (project PAIX, grant no. 1 403-073), Montpellier, France, for financial support. The Ecos Nord programme supported inter-laboratory mobility (grant no. C15A01). Leidy Rache was supported by Colciencias with a doctoral fellowship (call No. 528, 2011). Olivier Pruvost acknowledges the European Union (ERDF contract GURDT I2016‐1731‐0006632) and Réunion regional council for support."

Response: As suggested, we did not include any funding statement within the manuscript. Please use the current version of the funding statement for publication. 

Response: We included the original image of an agarose gel showing the amplification results for the multiplex PCR reactions in Supporting information “Fig. S4” 

7. We note that you have included the phrase “data not shown” in your manuscript. Unfortunately, this does not meet our data sharing requirements. PLOS does not permit references to inaccessible data. We require that authors provide all relevant data within the paper, Supporting Information files, or in an acceptable, public repository. Please add a citation to support this phrase or upload the data that corresponds with these findings to a stable repository (such as Figshare or Dryad) and provide and URLs, DOIs, or accession numbers that may be used to access these data. Or, if the data are not a core part of the research being presented in your study, we ask that you remove the phrase that refers to these data.

Response: Thanks for the suggestion, the phrase was removed because the data are not a core part of the research.

8. We note that you have referenced (ie. Rache L, Blondin L, Flores C, Trujillo C, Szurek B, Restrepo S, et al. [40]) which has currently not yet been accepted for publication. Please remove this from your References and amend this to state in the body of your manuscript: (ie “Rache L. et al. [Unpublished]”) as detailed online in our guide for authors

http://journals.plos.org/plosone/s/submission-guidelines#loc-reference-style "

Response: the paper is published already and can be accessed in this page: https://apsjournals.apsnet.org/doi/10.1094/PHYTO-06-18-0210-R. 

9. Please upload a new copy of Figure 3 as the detail is not clear. Please follow the link for more information:

https://blogs.plos.org/plos/2019/06/looking-good-tips-for-creating-your-plos-figures-graphics/

https://blogs.plos.org/plos/2019/06/looking-good-tips-for-creating-your-plos-figures-graphics/

Response: A new version of Figure 3 was uploaded. We hope that this new version meets the requirements. 

Response: The reference list was reviewed, this is complete and correct.

Reviewer: 

1) Line 171: « 3 loci in each multiplex » but in table 1: only 2 loci are present in Pool II. So one loci is missing in table 1, Pool II. This pool is correct in Figure 2. 

Response: The reviewer is correct, locus 2-29 was included in table 1. 

2) Line 168-170: primers were tested with DNA from 9 strains (4 from Columbia, 1 from Venezuela, 1 from Ivory Coast, 1 from Burkina, 2 from Mali). Lines 170-170: these strains are not all in the S1 table. Why? if they have been tested…

Response: The reviewer is correct, these strains are not in the S1 table because strains were used only to test the quality and design of the primers in agarose gels and they were not genotyped with labelled primers in a capillary electrophoresis system. Therefore, if we include them in table S1, they would not have comparable data. We included an extra sentence in this section: “These strains were only used to test the quality and design of the primers in agarose gels”. 

- Line 195: Xpm 2-20 is not in [40], it’s Xpm 1-20. Modify sentence

Response: The reviewer is correct, we included an extra sentence in the manuscript:(VNTR Xpm 1-20 nomenclature from [40]). Also, in table 2 was included “or Rache et al [40]”

- Line 237: « 11 loci were selected » but only 10 loci are in Table 2. Modify sentence

Response: “11 loci were selected” was replaced for “Ten loci were selected”

- Line 253-254 : « for loci Xpm1-29 and Xpm1-2 »: is it not rather Xpm2-29 and Xpm2-2 (from Table 2) ?.

Response: The reviewer is correct, we have now changed the numbers to “Xpm2-29 and Xpm2-2.

- Line 262-265 : it seem's that the names of the loci are wrong, please replace 1 with 2.

Response: The reviewer is correct, we have now changed the numbers in all instances.

- Line 311 : « Four VNTRs among the least discriminant of the MLVA-14 scheme were combined » : please name them in the sentence. 

Response: the sentence “Four VNTRs (2-31,2- 6, 2-7, 2-38) among the least discriminant of the MLVA-14 scheme were combined to the eight VNTRs (2-22, 2-18, 2-35, 2-20, 2-3, 2-23, 2-5, 2-33)…” was included

- Table 2: please explain what means the number between brackets after the motif sequence and why some letters are in bold for Xpm 2-2

Response: The explanation was included “number of times that motif sequence is present”. Letters in bold were a mistake in format. This mistake was corrected eliminating bold.

Reviewer: 

The authors could add a figure including disease symptoms on resistant, moderately resistant, and moderately susceptible cultivars.

Response: Thanks for the suggestion, the Figure (S3 Fig) was included.

---

## [Editor Report · Decision Letter 1]

25 Apr 2023

A minisatellite-based MLVA for deciphering the global epidemiology of the bacterial cassava pathogen Xanthomonas phaseoli pv. manihotis

PONE-D-22-33979R1

Dear Dr,

We’re pleased to inform you that your manuscript has been judged scientifically suitable for publication and will be formally accepted for publication once it meets all outstanding technical requirements.

Kind regards,

Tushar Shaw

Academic Editor

PLOS ONE
---

## [Editor Report · Acceptance letter]

2 May 2023

PONE-D-22-33979R1 

A minisatellite-based MLVA for deciphering the global epidemiology of the bacterial cassava pathogen Xanthomonas phaseoli<i> pv. manihotis 

Dear Dr. Bernal:

I'm pleased to inform you that your manuscript has been deemed suitable for publication in PLOS ONE. Congratulations! Your manuscript is now with our production department. 

Kind regards, 

on behalf of

Dr. Tushar Shaw 

Academic Editor

PLOS ONE